# Alcohol Use and Sexual Violence among Nursing Students in Catalonia, Spain: A Multicentre Cross-Sectional Study

**DOI:** 10.3390/ijerph18116036

**Published:** 2021-06-04

**Authors:** Concepció Fuentes-Pumarola, Zaira Reyes-Amargant, Alba Berenguer-Simon, David Ballester-Ferrando, Maria Dolors Burjalés-Martí, Assumpta Rigol-Cuadra, Elena Maestre-González, Carolina Rascón-Hernán, Paola Galbany-Estragués, Dolors Rodríguez-Martín

**Affiliations:** 1Health and Healthcare Research Group, Department of Nursing, University of Girona, 17003 Girona, Spain; concepcio.fuentes@udg.edu (C.F.-P.); alba.berenguer@udg.edu (A.B.-S.); carolina.rascon@udg.edu (C.R.-H.); 2Primary Care, Health Care Institute, 17005 Girona, Spain; zairareyes.94@gmail.com; 3Nursing Department, Faculty of Nursing, Rovira and Virgili University, Avinguda Catalunya, 35, 43002 Tarragona, Spain; mdolors.burjales@urv.cat; 4Department of Public Health, Mental Health and Perinatal Nursing, Faculty of Medicine, University of Barcelona, Feixa Llarga, s/n, 08907 L’Hospitalet del Llobregat, Spain; arigol@ub.edu (A.R.-C.); dolorsrodriguezmart@ub.edu (D.R.-M.); 5Department of Fundamental Care and Medical Surgital Nursing, Faculty of Medicine and Health Sciences, School of Nursing, Consolidated Research Group on Gender, Identity and Diversity (2017-SGR-1091), University of Barcelona, 08036 Barcelona, Spain; emaestre@ub.edu; 6Research Group on Methodology, Methods, Models and Outcomes of Health and Social Sciences (M3O), Faculty of Health Sciences and Welfare, Centre for Health and Social Care Research (CESS), University of Vic-Central University of Catalonia (UVic-UCC), C. Sagrada Família, 7, 08500 Vic, Spain; Paola.Galbany@uvic.cat

**Keywords:** sexual violence, sexual assault, alcohol, leisure, nursing students

## Abstract

(1) Background: Sexual violence (SV) has become common in universities for reasons related to unwanted social/peer pressures regarding alcohol/drug use and sexual activities. Objectives: To identify perceptions of SV and alcohol use and estimate prevalence among nursing students in Catalonia, Spain. (2) Methods: Observational descriptive cross-sectional study of a convenience sample of nursing students attending public universities. (3) Results: We recruited 686 students (86.11% women), who reported as follows: 68.7% had consumed alcohol, 65.6% had been drunk at least once in the previous year, 62.65% had experienced blackouts and 25.55% had felt pressured to consume alcohol. Drunkenness and blackouts were related (*p* < 0.000). Of the 15.6% of respondents who had experienced SV, 47.7% experienced SV while under the influence of alcohol and were insufficiently alert to stop what was happening, while 3.06% reported rape. SV was more likely to be experienced by women (OR: 2.770; CI 95%: 1.229–6.242; *p* = 0.014), individuals reporting a drunk episode in the previous year (OR: 2.839; 95% CI: 1.551–5.197; *p* = 0.001) and individuals pressured to consume alcohol (OR: 2.091; 95% CI: 1.332–3.281; *p* = 0.001). (4) Conclusions: Nursing instructors need to raise student awareness of both the effects of alcohol use and SV, so as to equip these future health professionals with the knowledge and skills necessary to deal with SV among young people.

## 1. Introduction

Sexual violence (SV) is one of the most frequent forms of gender violence and a major public health problem worldwide [1]. The World Health Organization (WHO) defines SV as “any sexual act, attempt to obtain a sexual act, or other act directed against a person’s sexuality using coercion, by any person regardless of their relationship to the victim, in any setting. It includes rape, defined as the physically forced or otherwise coerced penetration of the vulva or anus with a penis, other body part or object” [2]. This definition of SV includes sexual harassment, abuse and assault. While most of these acts are directed by men against women, men also experience SV [2,3].

According to WHO estimates [2], 1 in 3 women aged 15–45 years worldwide has experienced some form of sexual or physical violence in their lives. More specifically, between 0.3% and 12% of women have reported an experience of non-partner SV from the age of 15 years [4]. SV-related incidents are difficult to quantify because many cases are not reported and no fully reliable record-keeping system exists [5,6]. A Latin American study estimates that only 5% of adult victims of SV notify the police [7]. SV incidence is also higher in low-income countries in Africa and Latin America, although several studies conducted in the USA, a high-income country, have reported widespread sexual harassment, abuse and aggression in universities [8]. A study of SV for the period 1995–2013 by the US Department of Justice [9] concluded that women in the 18–24 age bracket (the typical age of attendance at university) were at greater risk of experiencing some form of SV than women in other age brackets. Alarming rates of SV among university students were documented for the first time 30 years ago and trends have not improved; indeed, the US Department of Justice reported in 2015 that incidence had increased by 10% over the previous decade [10,11].

Alcohol and drug use is frequent in university settings, where SV typically occurs for reasons related to social coercion, i.e., the feeling that to be socially accepted one has to give in to peer pressure to consume alcohol/drugs and engage in sexual activities, even if these make a person feel uncomfortable [12]. A study conducted at a university in the US northwest reported that 15% of first-year women were victims of rape when incapacitated by alcohol/drugs [13]; the explanation is that parties in universities take place in permissive environments where alcohol/drugs are freely available and affordable [14]. 

In relation to alcohol, several studies have linked risk behaviours and sexual relations in leisure settings with its consumption, as alcohol is often used to increase the chances of a sexual encounter or to modify a person’s behaviour or will, so they become more amenable to sex [15,16]. Alcohol intake is the most important risk factor for non-consensual sexual relations and forced touching [17,18], although its consumption affects men and women differently: in women, the capacity to react to alarm signals is reduced, i.e., protective behavioural strategies are undermined [19], whereas in men, impulses are disinhibited and aggressiveness increases [20]. Alcohol also heightens sexual arousal and desire, in turn increasing interest in sex and fostering other risky behaviours [21,22,23]. Drug-facilitated sexual assault (DFSA) is a term used to describe sexual intercourse with someone who is partially or fully incapacitated (and so unable to grant consent) due to the effects of the involuntary consumption of alcohol/drugs [24]. While different substances are associated with DFSA, ethyl alcohol is most commonly used [25]. Around three quarters (72%) of sexual assaults occur when the victim—usually a woman—is intoxicated and so unable to control or stop what is happening [26,27].

While evidence reported in numerous US and Canadian studies correlates alcohol intake with an increase in SV in leisure settings, few studies on this topic have been conducted among university students in Spain or, more specifically, among nursing students.

In Spain in 2016, there were 18.79 cases of SV per 100,000 inhabitants: 2.69 rapes and 16.10 cases of sexual abuse [28]. In the Spanish autonomous region of Catalonia in 2016, 20.9% of cases of SV against women by men who were not their partners took place in leisure contexts [29]. In Spain, alcohol is intimately associated with leisure among most young people. Alcohol is reported to be consumed regularly by young people aged 20–24 years [30] and, in 2017, 75.6% of students aged 14–18 years said they had consumed alcohol in the previous year [31].

The objectives of this study were, for a sample of nursing students in Spain, to identify perceptions of SV and alcohol use, to estimate SV prevalence in leisure settings and to analyse possible associations between SV, drunk episodes, blackouts and being pressured to consume alcohol.

## 2. Materials and Methods

### 2.1. Study Design

This cross-sectional descriptive non-randomized study, based on a self-administered questionnaire, was carried out between September 2017 and June 2018.

### 2.2. Population and Setting

The study population was 2471 nursing students attending four public universities in Catalonia (Autonomous University of Barcelona, University of Barcelona, University of Girona and Rovira i Virgili University in Tarragona).

### 2.3. Study Sample

The sample, taking into account the 20% prevalence rate for sexual abuse and assault reported in other studies conducted in Spain [32,33,34], was calculated on the basis of a 95% confidence level, a 3% precision level and a design effect of 1, resulting in an estimated sample size of *n* = 626. Inclusion criteria were to be present in the classroom on the day of data collection and to complete the questionnaire. A total of 686 participants were recruited.

### 2.4. Instruments

The survey was based on an ad hoc questionnaire in two parts. The first part collected (a) demographic data, namely, age, gender (respondents self-identified as a man or woman) and academic year (first to fourth year) and (b) data on alcohol use (“Do you consume alcohol in leisure settings?”), drunk episodes in the previous year (“In the last year, how often have you been drunk?”), blackouts in the previous year (“Have you had a blackout—failed to remember what happened during several hours—in the last year?”) and pressures to consume alcohol (“Have you ever felt under pressure to take alcohol?”). The second part reflected reported experiences of sexual abuse and sexual assault. The questionnaire was based on the Sexual Experiences Survey-Short Form Victimization (SES-SFV) instrument developed by Koss et al. [35,36], which addresses four different experiences of SV as follows: (1) “Someone fondled, kissed, or rubbed up against the private areas of my body (lips, breast/chest, crotch or butt) or removed some of my clothes without my consent (but did not attempt sexual penetration)”; (2) “Someone had oral sex with me or made me have oral sex with them without my consent”; (3) “A man put his penis into my vagina, or someone inserted fingers or objects without my consent”; (4) “A man put his penis into my butt, or someone inserted fingers or objects without my consent”.

Students were also asked to indicate the frequency of SV experiences since starting university, whether their experiences had occurred in a leisure context, whether they had been under the influence of alcohol to the point that they were insufficiently aware to stop what was happening and whether, even though nothing happened, someone had attempted something. Also explored was if they or someone they knew had been raped.

### 2.5. Data Collection

Nursing faculties were contacted to agree a day and time to administer the questionnaires to first-, second-, third- and fourth-year nursing students in situ in their classrooms.

Researchers visited the classrooms to inform the students of the objectives and purposes of the study, to recruit volunteers and to administer the questionnaire. Female researchers informed the participants of the importance of the study, that the anonymity of their responses was guaranteed and that they could at any time notify their withdrawal. Given the sensitivity of the topic, participants were asked to be sincere in their responses. The questionnaire was administered in electronic format using a QR code, while a paper format was also available to participants in case of technical problems or for participants who felt this format better assured their confidentiality. The estimated time required to complete the questionnaire was 10 min.

### 2.6. Ethical Considerations

The study was approved by the ethics committee of the University of Barcelona. Permission to carry out the study was obtained from nursing faculty deans and teachers were duly informed. Students who participated in the study were informed about the study, were guaranteed data confidentiality and anonymization in accordance with Spanish legislation on personal data protection [37] and gave their signed informed consent.

### 2.7. Data Analysis

A univariate descriptive analysis was conducted of all variables; continuous variables were described in terms of mean and standard deviation (SD) and median and interquartile range (IQR), while categorical variables were reported as percentages. In a bivariate analysis, associations between categorical variables were tested using Pearson’s chi-square test or Fisher’s test, while logistic regression analysis was performed with variables that proved significant in the bivariate analysis. Results were considered statistically significant for *p* < 0.05 and a confidence interval (CI) of 95%. Statistical analyses were performed using SPSS for Windows, version 23 (SPSS, Chicago, IL, USA).

## 3. Results

### 3.1. Sample Characteristics

The sample was composed of *n* = 686 nursing students, 86.11% (*n* = 591) of whom self-reported as women and 13.9% (*n* = 95) as men. Mean (SD) age was 21.36 (4.14) years and median (IQR) age was 20 (3). By academic year, first-, second-, third- and fourth-year students accounted for 31.9%, 28.75%, 23.89% and 17.36% of the sample, respectively.

### 3.2. Alcohol Use and Drunk Episodes

Most students (84.58%; *n* = 576) consumed alcohol always, almost always or sometimes when they went out at night, while 15.42% (*n* = 105) never or almost never consumed alcohol. No differences were found by gender or academic year (*p* = 0.861 and *p* = 0.102, respectively).

Two thirds of the respondents (65.6%; *n* = 448) reported having experienced drunk episodes at least once in the previous year, while 12% (*n* = 82) reported having this experience more than 10 times. Differences by gender and academic year were not statistically significant (*p* = 0.114 and *p* = 0.192, respectively).

### 3.3. Pressures to Consume Alcohol

Around a quarter of the respondents (25.55%; *n* = 174) reported feeling pressured to consume alcohol in leisure settings. There were no statistically significant differences by gender (*p* = 0.817) or academic year (*p* = 0.151).

Of the respondents pressured to consume alcohol, 66.09% reported having experienced a drunk episode at least once in the previous year. No statistically significant differences were observed between these students and those who had not experienced pressures (Table 1).

### 3.4. Blackout Experiences

Almost two thirds of the respondents (62.65%; *n* = 425) declared having experienced a blackout, with no significant differences by gender (*p* = 0.589). By academic year there were statistically significant differences in blackout experiences, with increased percentages for second-, third- and fourth-year students (*p* < 0.000) (Table 2).

Of the respondents who reported a drunk episode in the previous year, 78.03% (*n* = 348) also declared having experienced a blackout, compared to 21.97% (*n* = 98) who reported drunk episodes but no blackouts (*p* = 0.000).

### 3.5. SV Experiences

Of the 686 nursing students in our sample (Table 3), 15.6% (*n* = 107) stated that they had experienced some type of SV (women 93.5% (*n* = 100) versus men 6.5% (*n* = 7); *p* = 0.017). Of the students who had experienced some type of SV, 47.7% (*n* = 51) experienced SV while under the influence of alcohol and so were not sufficiently alert to stop what was happening (women 92.2% (*n* = 47) versus men 7.8% (*n* = 4); *p* = 0.603). Most of those experiences, therefore, primarily affected women and most (72.9%; *n* = 78) occurred in a leisure context.

A total of 21 (3.06%) respondents—2.2% of the women (*n* = 19) and 3.3% of the men (*n* = 2)—confirmed that they had been raped, while 104 (15.16%) respondents—15.6% of the women (*n* = 90) and 14.9% of the men (*n* = 14 men)—reported that they knew someone who had been raped. Results by gender and academic year were not statistically significant (*p* = 0.556 and *p* = 0.640, respectively).

Of the respondents who said they had been raped, equal proportions said they had experienced drunk episodes or blackouts in the previous year (87.71%) and over half (51.14%) had been pressured to consume alcohol; differences in relation to respondents who had not experienced rape were statistically significant (Table 4).

A logistic regression analysis adjusted by gender and specific variables as identified in the bivariate analysis showed that the following were more likely to experience SV in SES-SFV terms: women who reported a drunk episode or a blackout in the previous year or who had been pressured to consume alcohol (Table 5).

## 4. Discussion

Krebs et al. [38] reported a relationship between leisure, alcohol use and SV. In our study of university nursing students aimed at investigating this possible relationship, we found that women were more likely to have experienced SV in leisure settings if they were pressured to consume alcohol and if they reported drunk and blackout episodes in the previous year.

Corroborating a similar finding reported for the French university population [39], 68.57% of the nursing students in our sample said they consumed alcohol in leisure settings (with similar distributions by gender and academic year). Other European studies have reported lower levels of alcohol consumption; in the UK, high-risk alcohol consumption was observed in 55.6% of nursing students [40], while in Germany, approximately 60% of university students reported alcohol consumption at least once a week, with higher consumption in men than in women [41]. A study conducted in Chile reported higher alcohol consumption by fourth-year students [42]. The higher alcohol consumption reported in our study may be due to cultural differences; teenagers in Spain begin to drink alcohol between 14 and 18 years of age, for an incidence rate in 2016 of 47.7% [31].

Detecting cases of DFSA is frequently difficult due to the lack of evidence and victim recall problems. One alert, however, is blackout [43,44]. Of the students in our sample, 62.65% reported having experienced a blackout—a percentage higher than reported by other authors—while 78.30% of those who reported a drunk episode in the previous year declared having experienced a blackout. One study found that although women consumed less alcohol, those who reported recent blackouts were more likely to have had to deal with unsolicited sexual advances [45].

In terms of the situations covered by the SES-SFV instrument, 15.6% of our respondents said they had experienced an episode of SV. Higher percentages of SV have been reported by other studies. Untied et al. [46] reported that 37% of women reported a history of unwanted sexual contact, attempted rape or sexual coercion and that 8.4% reported rape. In our study, 3.06% of the students reported rape, while 3.4% and 1.46% reported that “A man put his penis into my vagina, or someone inserted fingers or objects without my consent” and that “A man put his penis into my butt, or someone inserted fingers or objects without my consent”, respectively. Other authors report rates of 22.7% [47] and 28% [48] for unwanted sexual experiences at university. Carey et al. [13] reported that 15.4% of the first-year university students in their study reported ‘incapacitated rape’ (their term for DFSA). In our study, 7.4% of the students reported having experienced some form of SV while under the influence of alcohol and being insufficiently alert to stop what was happening. As noted by other authors, there appears to be a link between drunk episodes/blackouts and the probability of experiencing SV [48]. Our study reflects that same pattern: women who were drunk, experienced blackouts and felt pressured to consume alcohols were more likely to report SV.

In our study, the behavioural reasons for the consumption of alcohol, which happens to be easily accessed and affordable in Spain, have not been addressed. While several studies have found that pluralistic ignorance influences drinking behaviours [49], further research is needed into student opinions and motives regarding alcohol consumption, so as to investigate beliefs in relation to the effects of alcohol and sexual behaviours and develop and implement suitable interventions. Providing information on peer behaviours and attitudes has been reported to reduce alcohol consumption and foster more responsible behaviours [50], so such a strategy might, in the case of our students, be useful in fostering healthy and responsible sexual behaviours.

The limitations of this study include the fact that the truthfulness of responses to the survey cannot be assured, that clear cut cause-effect relationships cannot be established given the cross-sectional nature of the study and that our use of a convenience sample means that results cannot be extrapolated to the general student population. Our study, furthermore, did not identify lesbian, gay, bisexual, transgender and intersex (LGBTI) respondents (students could only choose to self-report as a man or woman). This issue needs to be addressed in future studies for Spain, as has been done for the USA. One study of 27 university campuses by the Association of American Universities [18] reported that while 23.1% of women and 5.4% of men had experienced some form of SV, more LGBTI individuals (21%) reported having experienced SV than non-LGBTI individuals (18%).

## 5. Conclusions

Although we report lower prevalence of SV compared to other studies, the number of SV cases reported gives cause for concern, especially in relation to pressures exerted regarding alcohol consumption. Nursing students—our health professionals of the future—need to be fully informed regarding SV and the impact of alcohol on SV. Indeed, academic curricula in general need to include training on the effects of alcohol and other drugs and actions aimed at preventing SV, while social and political leaders need to tackle this dual public health problem by implementing actions aimed at fostering health.

The consumption of alcohol, the association with blackouts and the relationship between alcohol consumption, leisure settings and SV constitute sufficient reason to implement changes in nurse training, to ensure that future nurses are equipped with the necessary knowledge and skills to become agents of societal change regarding attitudes and beliefs about SV among young people.

## Figures and Tables

**Table 1 ijerph-18-06036-t001:** Drunk episodes and pressures to consume alcohol (*n* = 678).

Drunk Episode (Previous Year)	Pressure YES *n* (%)	Pressure NO *n* (%)	*p*-Value *
YES	115 (66.09)	330 (65.48)	0.883
NO	59 (33.91)	174 (34.52)	
Total	174 (100)	504 (100)	

* Chi-square test: significance *p* < 0.05.

**Table 2 ijerph-18-06036-t002:** Blackouts by academic year (*n* = 668).

Academic Year				
Blackout	1st *n* (%)	2nd *n* (%)	3rd *n* (%)	4th *n* (%)
YES	112 (52.58)	109 (60.89)	121 (75.63)	76 (65.52)
NO	101 (47.42)	70 (39.11)	39 (24.38)	40 (34.48)
Total	213	179	160	116

Chi-square test: significance *p* < 0.05.

**Table 3 ijerph-18-06036-t003:** Prevalence of sexual violence by gender (*n* = 686).

		Experience * *n* (%)	Experience under the Influence of Alcohol *n* (%)	Even though It Didn’t Happen, Someone Tried *n* (%)	Even though It Didn’t Happen, Someone Tried When I Was under the Influence of Alcohol *n* (%)
Someone fondled, kissed, or rubbed up against the private areas of my body (lips, breast/chest, crotch or butt) or removed some of my clothes without my consent but did not attempt sexual penetration)	Women **	93.9 (92)	93.5 (43)		
	Men **	6.1 (6)	6.5 (3)		
	Total	14.2 (98)	6.9 (46)		
	*p*-value ***	0.094	0.212		
	More than once	32.65 (32)	23.9 (11)		
	Know someone who had the experience	33.58 (226)	46.46 (105)		
Someone had oral sex with me or made me have oral sex with them without my consent	Women **	87.5 (21)	80 (8)	85.3 (29)	88.2 (15)
	Men **	12.5 (3)	20 (2)	14.7 (5)	11.68 (2)
	Total	3.5 (24)	1.60 (10)	4.96 (34)	2.58 (17)
	*p*-value **	0.974	0.785	0.921	0.642
	More than once **	41.6 (0)	60 (6)	29.4 (10)	29.4 (5)
	Know someone who had the experience	8.13 (55)	47.27 (26)	6.17 (41)	41.46 (17)
A man put his penis into my vagina, or someone inserted fingers or objects without my consent	Women **	100 (20)	100 (9)	100 (22)	100 (8)
	Men **	0 (0)	0 (0)	(0)	0 (0)
	Total	3.4 (20)	1.65 (9)	3.18 (22)	1.48 (8)
	*p*-value ***	0.344	0.983	0.160	0.985
	More than once **	40 (8)	22.2 (2)	36.3 (8)	0.74 (4)
	Know someone who had the experience	12.02 (72)	54.16 (9)	6.42 (8)	63.15 (24)
A man put his penis into my butt, or someone inserted fingers or objects without my consent.	Women **	80 (8)	75 (3)	66.6 (8)	50 (2)
	Men **	20 (2)	25 (1)	33 (4)	50 (2)
	Total	1.46 (10)	0.64 (4)	1.75 (12)	0.64 (4)
	*p*-value ***	0.417	0.607	0.100	0.035
	More than once **	30 (3)	25 (1)	41.6 (5)	25 (1)
	Know someone who had the experience	2.80 (18)	55.55 (10)	2.35 (15)	53.55 (8)

* Respondents may have had the experience more than once. ** % with respect to the total of affirmative answers. *** Chi square test: significance *p* < 0.05.

**Table 4 ijerph-18-06036-t004:** Rape associated with drunk episodes, blackouts and pressures to consume (*n* = 668).

		Drunk Episodes		Pressures		Blackouts	
	Total *n* (%)	YES *n* (%)	NO *n* (%)	YES *n* (%)	NO *n* (%)	YES *n* (%)	NO *n* (%)
Rape YES	21 (3.05)	18 (87.71)	3 (14.30)	12 (51.14)	9 (42.86)	18 (87.71)	3 (14.30)
Rape NO	647 (96.95)	422 (65.20)	225 (34.80)	160 (24.73)	487 (75.27)	400 (61.92)	246 (38.10)
*p*-value *		0.051		0.001		0.027	

* Chi square test: significance *p* < 0.05.

**Table 5 ijerph-18-06036-t005:** Regression analysis of sexual violence: gender, drunk episodes, blackout episodes and pressures to consume alcohol.

	B	SE	Wald	df	Sig	Exp(B)	95% CI by Exp(B)	
							Lower	Higher
Gender *	1.019	0.414	6.042	1	0.014	2.770	1.229	6.242
Drunk	1.043	0.309	11.439	1	0.001	2.839	1.551	5.197
Blackouts	0.571	0.278	4.224	1	0.040	1.770	1.027	3.049
Pressures	0.738	0.230	10.292	1	0.001	2.091	1.332	3.281
Constant	−3.993	0.496	64.699	1	0.000	0.018		

* Gender: man 1, woman 2. B (beta coefficient); CI (confidence interval); df (degrees of freedom); Exp(B) (exponentiation of the B coefficient, odds ratio value); SE (standard error); sig (statistical significance).

## Data Availability

Data availability is currently restricted due to ongoing data collection and to maintain participant privacy.

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
