# Peer review of "Alcohol Use and Sexual Violence among Nursing Students in Catalonia, Spain: A Multicentre Cross-Sectional Study"

_ijerph, 2021, doi:10.3390/ijerph18116036_

Round 1
Reviewer 1 Report
Regarding Section 3.4 - Blackout experiences: The authors report that there are "statistically significant differences in blackout experiences in the increased percentages for second- and third-year students." However, the percentage of fourth students reporting blackouts is 65%, which is between the percentages for second year (60%) and third year (75%) students. Therefore, isn't the percentage of fourth year students reporting blackouts also statistically significant?
Section 3.5: It is hard to imagine that the difference in percentage of women vs. men who either have experienced rape, or know of someone who has, is not statistically significant. You may wish to reassess the statistics here.
In Table 3, it is unclear what is meant by the "Affirmative" row (as opposed to the male vs. female rows). This seems confusing since I would think the number who answered affirmatively would equal the total for male + female for a given statement.
The authors used the word "pressurized" when the accepted English word is actually "pressured".
Author Response
Point 1: Regarding Section 3.4 - Blackout experiences: The authors report that there are "statistically significant differences in blackout experiences in the increased percentages for second- and third-year students." However, the percentage of fourth students reporting blackouts is 65%, which is between the percentages for second year (60%) and third year (75%) students. Therefore, isn't the percentage of fourth year students reporting blackouts also statistically significant?
Response 1: Thank you for your observation. The text has been modified as follows: By academic year there were statistically significant differences in blackout experiences, with increased percentages for second-, third- and fourth-year students (p<0.000)
Point 2: Section 3.5: It is hard to imagine that the difference in percentage of women vs. men who either have experienced rape, or know of someone who has, is not statistically significant. You may wish to reassess the statistics here.
Response 2: There were no statistically significant differences between men and women, since the percentage of men who experienced rape was 2.2% and of women was 3.3%. The wording has been modified to show the percentages by sex and not the totals:
A total of 21 (3.06%) respondents – 2.2% of the women (n=19) and 3.3% of the men (n=2) – confirmed that they had been raped, while 104 (15.16%) respondents – 15.6% of the women (n=90) and 14.9% of the men (n=14 men) – reported that they knew someone who had been raped. Results by gender and academic year were not statistically significant (p=0.556 and p=0.640, respectively).
Point 3: In Table 3, it is unclear what is meant by the "Affirmative" row (as opposed to the male vs. female rows). This seems confusing since I would think the number who answered affirmatively would equal the total for male + female for a given statement.
Response 3: Following your indications Table 3 has been modified to make it easier to understand. That row has been renamed “Total” and has been placed after gender.
Point 4: The authors used the word "pressurized" when the accepted English word is actually "pressured".
Response 4: Corrected.
Reviewer 2 Report
This was a vast undertaking in terms of the sheer number of people the authors were able to get to respond to such sensitive questions. The topic is an important one but I do think that the data the authors tirelessly collected deserves a deeper theoretical framework.
As it stands, there are some implicit assumptions made about SV, such as the victim is the one who needs educating in order to prevent SV. It seems to be communicated that alcohol plus young people means that SV/sexual assault will happen. But, what are the mechanisms by which this occurs? The authors suggest from the outset that this data should be used to help nursing instructors better prepare their students to deal with patients that have experienced SV- but the way in which that preparation should unfold should be addressed. And it needs to be something more compelling than 'try to not get raped.' This is where a stronger theoretical grounding/framing would benefit the interpretation of the results.
For example, there are several social psychological concepts that I might suggest:
1) pluralistic ignorance has been shown in many different studies to influence drinking behavior in college students. This, along with the affordability of and accessibility to alcohol that the authors point out, could provide one way prepare students to be mindful of their own behavior (whether as a perpetrator or a target of SV while drinking). That is, educating students about the REAL numbers of students who drink, or highlighting the students who do not, could forearm them early in their post-secondary education. https://pubmed.ncbi.nlm.nih.gov/8433272/#:~:text=All%20studies%20found%20widespread%20evidence,with%20respect%20to%20one's%20friends.
2) Self-fulfilling prophecies can come into play when they are coupled with beliefs about alcohol's effects on behavior. Often it is the belief that alcohol makes people do things that they would not normally do, such as SV, that can lead to inhibition and risky behaviors.
3) Finally, where is the analysis of the behavior of the perpetrator in all of this? A companion study (as a follow up) that explores that would be required. The authors have done well with their demonstration that SV is prevalent and alcohol consumption is prevalent but an exploration of the mechanisms by which these instances occur is almost mandatory when writing about SV in 2021.
Thank you for taking on this task- I would look forward to seeing your take on the potential mechanisms for what you have observed.
Author Response
Point 1: 1) pluralistic ignorance has been shown in many different studies to influence drinking behavior in college students. This, along with the affordability of and accessibility to alcohol that the authors point out, could provide one way prepare students to be mindful of their own behavior (whether as a perpetrator or a target of SV while drinking). That is, educating students about the REAL numbers of students who drink, or highlighting the students who do not, could forearm them early in their post-secondary education. https://pubmed.ncbi.nlm.nih.gov/8433272/#:~:text=All%20studies%20found%20widespread%20evidence,with%20respect%20to%20one's%20friends.
Response 1: Thank you for this interesting insight. Our study was a descriptive quantitative study. We are currently, in fact, conducting a qualitative study to better explore student behaviours. We have included the following text in the discussion:
In our study, the behavioural reasons for the consumption of alcohol, which happens to be easily accessed and affordable in Spain, have not been addressed. While several studies have found that pluralistic ignorance influences drinking behaviours [50], further research is needed into student opinions and motives regarding alcohol consumption, so as to investigate beliefs in relation to the effects of alcohol and sexual behaviours and develop and implement suitable interventions. Providing information on peer behaviours and attitudes has been reported to reduce alcohol consumption and foster more responsible behaviours [51], so such a strategy might, in the case of our students, be useful in fostering healthy and responsible sexual behaviours.
Point 2: 2) Self-fulfilling prophecies can come into play when they are coupled with beliefs about alcohol's effects on behavior. Often it is the belief that alcohol makes people do things that they would not normally do, such as SV, that can lead to inhibition and risky behaviors.
Response 2: We fully agree, and this will be an aspect to explore in future research. In the manuscript it is already mentioned in the introduction how alcohol can affect behaviours, but it is also necessary to investigate beliefs that can lead to risky behaviours. For this reason, the paragraph reproduced in the response to point 1 has been included in the discussion.
Point 3: 3) Finally, where is the analysis of the behavior of the perpetrator in all of this? A companion study (as a follow up) that explores that would be required. The authors have done well with their demonstration that SV is prevalent and alcohol consumption is prevalent but an exploration of the mechanisms by which these instances occur is almost mandatory when writing about SV in 2021.
Response 3: The behaviour of perpetrators has not been addressed, becuase our descriptive study was concerned with prevalence. We are currently conducting a qualitative study with students from different universities to explore this issue.
Reviewer 3 Report
“Nursing students and alcohol use in leisure settings: A multi-centric cross-sectional study of sexual violence”
- I suggest that authors change the title to provide more clarity: “Alcohol use and Sexual Violence among Nursing Students in Catalonia, Spain: a Multi-centric cross-sectional study”
- Line 19 - 21 Objectives: To identify perceptions of SV and alcohol use and estimate prevalence among nursing students in Spain: this was not a representative study in Spain, I suggest you replace Spain with Catalonia, Spain, and representative with convenience sample.
- Line 22 - Results: “Recruited were 686 students” this sounds funny, please revise English
- Line 25 – “Of the 15.6% of respondents who had experienced SV, 54.68% were drunk and unaware of what was happening”: if they were unaware of what was happening because they were so heavily drunk, how can they recall and affirm that they experienced SV?
- Please include information on why it is relevant to explore these variables among nursing students.
- Please include epidemiologic data on alcohol consumption in Spain, if available.
- Lines 90-91: Please provide more information on the objectives to accommodate what was conducted during data analysis.
- Please inform if this was a randomized study or not, if not, generalization of results is impossible. It seems like this is not a representative sample.
- Lines 95-96: please provide further details.
- Replace sex with gender across the paper.
- Data collection: authors must provide further details on how data was collected.
- Please include a table to communicate your logistic regression results.
- Please provide more explanation on why nursing student’s alcohol consumption is high (e.g., stress coping, peer pressure, cultural setting, etc)
- Please include implications of your results expanding the idea that Nursing instructors need to raise student awareness of both the effects of alcohol use and SV, as well as policy makers and social decisors.
Best wishes.
Author Response
“Nursing students and alcohol use in leisure settings: A multi-centric cross-sectional study of sexual violence”
Point 1: I suggest that authors change the title to provide more clarity: “Alcohol use and Sexual Violence among Nursing Students in Catalonia, Spain: a Multi-centric cross-sectional study”
Response 1: Thank you. The title has been changed as suggested.
Alcohol Use and Sexual Violence among Nursing Students in Catalonia, Spain: a Multicentre Cross-sectional Study
Point 2: Line 19 - 21 Objectives: To identify perceptions of SV and alcohol use and estimate prevalence among nursing students in Spain: this was not a representative study in Spain, I suggest you replace Spain with Catalonia, Spain, and representative with convenience sample.
Response 2: We have made this modification in the abstract and text as follows:
To identify perceptions of SV and alcohol use and estimate prevalence among nursing students in Catalonia, Spain. (2) Methods: Observational descriptive cross-sectional study of a convenience sample of nursing students attending public universities
Point 3: Line 22 - Results: “Recruited were 686 students” this sounds funny, please revise English.
Response 3: This was to avoid starting a sentence with a number and to keep the sentence subject (students) near the relative pronoun (who). However, we have found a workaround, as follows:
We recruited 686 students
Point 4:“Of the 15.6% of respondents who had experienced SV, 54.68% were drunk and unaware of what was happening”:if they were unaware of what was happening because they were so heavily drunk, how can they recall and affirm that they experienced SV?
Response 4: We have modified this sentence, which we agree is illogical, and have also corrected the percentage in the abstract that did not correspond to the results:
47.7% experienced SV while under the influence of alcohol and insufficiently alert to stop what was happening
Point 5: Please include information on why it is relevant to explore these variables among nursing students.
Response 5: Alcohol use and SV among nursing students was studied, as no evidence has been reported for this specific group of health science students, as indicated at the end of the introduction:
While evidence reported in numerous US and Canadian studies correlates alcohol intake with an increase in SV in leisure settings, few studies on this topic have been conducted among university students in Spain or, more specifically, among nursing students
Point 6: Please include epidemiologic data on alcohol consumption in Spain, if available.
Response 6: The end of the introduction includes data on alcohol consumption by young people in Spain:
In Spain in 2016, there were 18.79 cases of SV per 100,000 inhabitants: 2.69 rapes and 16.10 cases of sexual abuse (Eurostat, 2016). In the Spanish autonomous region of Catalonia in 2016, 20.9% of cases of SV against women by men not their partners took place in leisure contexts [28]. In Spain, alcohol is intimately associated with leisure among most young people. Alcohol is reported to be consumed regularly by young people aged 20-24 years [29] and, in 2017, 75.6% of students aged 14-18 years said they had consumed alcohol in the previous year [30]
Point 7: Lines 90-91: Please provide more information on the objectives to accommodate what was conducted during data analysis.
Response 7: The wording of the objectives has been modified:
The objectives of this study were, for a sample of nursing students in Spain, to identify perceptions of SV and alcohol use, to estimate SV prevalence in leisure settings, and to analyse possible associations between SV, drunk episodes, blackouts and being pressured to consume alcohol.
Point 8: Please inform if this was a randomized study or not, if not, generalization of results is impossible. It seems like this is not a representative sample.
Response 8: We have mentioned in Methods that the study is non-randomized and have modified the limitations as follows:
This cross-sectional descriptive non-randomized study, based on a self-administered questionnaire, was carried out between September 2017 and June 2018.
The limitations of this study include the fact that the truthfulness of responses to the survey cannot be assured, that clearcut cause-effect relationships cannot be established given the cross-sectional nature of the study and that our use of a convenience sample means that results cannot be extrapolated to the general student population.
Point 9: Lines 95-96: please provide further details.
Response 9: Section 2 has been expanded and 2.1 has been included:
This cross-sectional descriptive non-randomized study, based on a self-administered questionnaire, was carried out between September 2017 and June 2018.
Point 10: Replace sex with gender across the paper.
Response 10: Done. Please note that we have included information to indicate that the questionnaire respondents self-reported as a man or woman.
Point 11: Data collection: authors must provide further details on how data was collected.
Response 11: The text now includes details as follows:
Female researchers informed the participants of the importance of the study, that the anonymity of their responses was guaranteed, and that they could at any time notify their withdrawal. Given the sensitivity of the topic, participants were asked to be sincere in their responses. The questionnaire was administered in electronic format using a QR code, while a paper format was also available to participants in case of technical problems or for participants who felt this format better assured their confidentiality. The estimated time required to complete the questionnaire was 10 minutes
Point 12: Please include a table to communicate your logistic regression results.
Response 12: The text has been modified and a table (Table 5) has been included.
Point 13: Please provide more explanation on why nursing student’s alcohol consumption is high (e.g., stress coping, peer pressure, cultural setting, etc)
Response 13: Our study is a descriptive quantitative study, and we are currently conducting a qualitative study to better explore student behaviours and motivations. We have also included the following text in the discussion:
In our study, the behavioural reasons for the consumption of alcohol, which happens to be easily accessed and affordable in Spain, have not been addressed. While several studies have found that pluralistic ignorance influences drinking behaviours [50], further research is needed into student opinions and motives regarding alcohol consumption, so as to investigate beliefs in relation to the effects of alcohol and sexual behaviours and develop and implement suitable interventions. Providing information on peer behaviours and attitudes has been reported to reduce alcohol consumption and foster more responsible behaviours [51], so such a strategy might, in the case of our students, be useful in fostering healthy and responsible sexual behaviours.
The discussion already includes:
The higher alcohol consumption reported in our study may be due to cultural differences; teenagers in Spain begin to drink alcohol between 14 and 18 years of age, for an incidence rate in 2016 of 47.7% [42].
Point 14: Please include implications of your results expanding the idea that Nursing instructors need to raise student awareness of both the effects of alcohol use and SV, as well as policy makers and social decisors.
Response 14: Included in the conclusions is the following:
Indeed, academic curricula in general need to include training on the effects of alcohol and other drugs and actions aimed at preventing SV, while social and political leaders need to tackle this dual public health problem by implementing actions aimed at fostering health.
Round 2
Reviewer 2 Report
Thank you for your responses- I look forward to the follow up study that you are currently working on.
Reviewer 3 Report
Thank you for the updates. The overall quality of the articles has very much improved.